# Predictors of burnout among academic family medicine faculty: Looking back to plan forward

Viola Antao *, Paul Krueger, Christopher Meaney, Jeffrey C. Kwong, David White

Department of Family and Community Medicine, University of Toronto, Toronto, Ontario, Canada

☙ These authors contributed equally to this work.
* viola.antao@utoronto.ca

## Abstract

### Objective

To identify the prevalence and predictors of burnout among academic family medicine faculty.

### Design

A comprehensive survey of academic family medicine faculty on burnout, perceptions of work life, and practice in 2011.

### Setting

A large, distributed Department of Family and Community Medicine at the University of Toronto.

### Participants

All 1029 faculty members were invited to participate.

### Main outcome measures

Maslach Burnout Inventory three subscales (emotional exhaustion, depersonalization, personal accomplishment).

### Results

The survey response rate was 66.8% (687/1029). The prevalence of high emotional exhaustion scores was 27.0% and high depersonalization was 9.2%, whereas the prevalence of high personal accomplishment scores was 99.4%. Bivariate analyses identified 27 variables associated with emotional exhaustion and 18 variables associated with depersonalization, including: ratings of the practice setting; leadership and mentorship experiences; job satisfaction; health status; and demographic variables. Multivariate analyses found four predictors of emotional exhaustion: lower ratings

**Data availability statement:** We are unable to make the dataset publicly available. At the time of participant recruitment, informed consent did not include provisions for data sharing outside of the research team. This restriction was also approved by our institution's Research Ethics Board (REB), which prohibits the public release of the dataset in its current form. Furthermore, the dataset contains highly sensitive information, including participant perceptions of both local and central leadership within their organizations. Given the context and potential for re-identification—particularly in small or closely connected professional settings—sharing the data could pose serious risks to participant confidentiality and could potentially lead to interpersonal or workplace conflicts. We remain committed to transparency and reproducibility and are happy to provide additional methodological details or, where appropriate, consider sharing de-identified data subsets under strict access conditions (e.g., through a data use agreement) if feasible within the bounds of our REB approval. Requests for access to data are subject to approval. Requests for data from eligible researchers may be dispatched to the following: Research Oversight & Compliance Human Research Ethics Unit - General Enquiries ethics.review@utoronto.ca (416) 946-3273.

**Funding:** The author(s) received no specific funding for this work.

**Competing interests:** The authors have declared that no competing interests exist.

of job satisfaction, poorer ratings of workplace quality, working ≥50 hrs/week, and poorer ratings of health status. Predictors of depersonalization included lower ratings of job satisfaction, ≤5 years in practice, lower ratings of health status, and poor ratings of mentorship received.

## Conclusions

This study describes the prevalence and predictors of burnout among physicians prior to the COVID-19 pandemic. Predictors that are potentially modifiable at local practice and systems levels include job satisfaction, workplace quality, hours worked, and mentorship received. New family physicians (≤5 years in practice) were at increased risk of depersonalization; strategies specific to this group may limit burnout and address the healthcare workforce crisis. Periodic studies are recommended to identify the impact of strategies implemented, emergent predictors, trends, and mitigating factors associated with burnout. The current crisis in family medicine indicates an urgent need to look back and plan forward.

## Introduction

Burnout is contributing to the ongoing depletion of the family physician workforce, exacerbating the crisis in primary care [1–3]. In Canada, an estimated 6.5 million adults lack access to primary care [4] yet 100 family medicine residency positions remained unfilled in 2023 [5]. Burnout among family physicians at various career stages and practice settings contributes to decreased interest in providing comprehensive care, increased intentions to leave practice, physician turnover, and early retirement [6,7].

Burnout is "a work-related syndrome of emotional exhaustion, depersonalization, and reduced feelings of personal accomplishment" first described in the 1970s [8,9]. Burnout remains prevalent. Despite extensive research and implementation of "wellness and engagement strategies" focused on physician resilience, a cogent understanding of causes and potential solutions is elusive [10,11]. Burnout may contribute to reduced quality of patient care [12], and significant health system costs [13], supporting the call to improve the work life of health care providers to optimize health system performance [14,15,17].

Physicians practicing family medicine have higher rates of burnout compared to physicians in other specialities [16,17]. Among family physicians, the prevalence ranged from 25–60% [18]. Studies have identified higher prevalence among those earlier in practice (<10 years) [17] and females [19,20]. Large debt, high clinical load, and childcare responsibilities are potential contributors to burnout in new family physicians [19,21].

In 2011, the Department of Family and Community Medicine (DFCM) at the University of Toronto conducted a "Faculty Work and Leadership Survey" to assess the quality of work life and leadership development for faculty [22–24]. The purpose of this study was to determine the prevalence and predictors of burnout among

academic family medicine faculty, given the limited literature specific to this group. With the growing workforce crisis in primary care, these findings can help to fill the historic gap in the literature, provide a comparative level or baseline for burnout, inform current efforts to mitigate burnout, and plan future research.

## Methods

### Setting

In 2011, the DFCM comprised 1029 faculty distributed across 14 family medicine teaching units and numerous community-based practices.

### Questionnaire

From September 26 – October 24, 2011, we conducted a web-based survey of all DFCM faculty, initiating with a recruitment email. The questionnaire collected information on burnout, perceptions of work life, practice, and demographic information. Consent was implied as participants were informed of the voluntary survey and could agree or refuse to participate. The questionnaire content, development, survey promotion, and implementation have been described in detail elsewhere [23]. Participants were informed about the survey

### Outcome measure

We used the Maslach Burnout Inventory (MBI) [25], a validated instrument for measuring burnout. It consists of 3 subscales: emotional exhaustion, a measure of feeling overextended by work; depersonalization, a measure of unfeeling and impersonal response toward care recipients; and personal accomplishment, a measure of feelings of efficacy and successful achievement in work. Each subscale (outcome) has specific cut-points (Table 1).

### Statistical analysis

For the bivariate analyses, we dichotomized the three cut points for emotional exhaustion, depersonalization, and personal accomplishment into "high" versus "low + moderate". We used chi-square tests to identify potential predictors of each outcome. We included statistically significant variables from the bivariate analyses in multivariable logistic regression analyses to identify parsimonious sets of predictors for each of the outcomes. Variables that were highly correlated or alternative ways to measure the same construct were excluded from the regression models to avoid multicollinearity. We report adjusted odds ratios and corresponding 95% confidence intervals. Goodness-of-fit of the final logistic regression models was assessed using various statistical techniques including the rho-square statistic [26].

### Research ethics

Ethics approval for this study was obtained from the University of Toronto Research Ethics Board (UTREB #00026748).

**Table 1. Cut points for determining low, moderate, and high emotional exhaustion, depersonalization, and personal accomplishment.**

| Likelihood of burnout | Scores | | |
|---|---|---|---|
| | Emotional Exhaustion | Depersonalization | Personal Accomplishment* |
| Low | 0-16 | 0-8 | 39-56 |
| Moderate | 17-26 | 9-13 | 32-38 |
| High | 27-63 | 14-35 | 0-31 |

*Personal accomplishment is scored in the opposite direction to emotional exhaustion and depersonalization such that lower scores indicate less personal accomplishment and higher likelihood of burnout.

 

## Results

### Participant characteristics

Respondents' mean age was 47 years (range 29–82 years); 52% were women; 87% were married or living with a partner; 72% identified as being from a white cultural background; and 76% were Canadian-born. Forty percent of participants reported working at their current site for ≤5 years, 30% for 6–15 years, and 30% for ≥16 years. Overall, faculty members worked on average 46 hours/week, with 88% having on-call duties.

### *Prevalence of emotional exhaustion, depersonalization, and personal accomplishment*

Of the 687 respondents, 623 (90.7%) completed the MBI questions. Table 2 describes the percentage of respondents reporting low, moderate, and high levels on the three MBI subscales.

### Bivariate analysis

Bivariate analyses were conducted for two of the subscales, emotional exhaustion and depersonalization. We did not analyze the personal accomplishment subscale due to lack of variation in that outcome, as almost all respondents (99%) scored high on personal accomplishment, similar to other studies [27]. Of the 27 statistically significant predictors of emotional exhaustion, 11 were faculty ratings of their local department, one was related to their main practice, two were leadership and mentorship experience variables, four were related to job satisfaction, six were related to health status variables, and three were related to demographic and practice variables (Table 3).

Of the 18 statistically significant predictors of depersonalization, eight were faculty ratings of their local department, one was related to mentorship experience, four were related to job satisfaction, three were related to health status variables, and two were related to demographic and practice variables (Table 4).

### Multivariable analyses

The logistic regression model for emotional exhaustion identified lower ratings of job satisfaction, poorer ratings of work-place quality, working ≥50 hours per week, and poorer ratings of health status as predictors of emotional exhaustion (Table 5).

The logistic regression model for depersonalization identified lower ratings of job satisfaction, shorter duration in practice, lower ratings of health status, and poorer ratings of mentorship received as predictors of depersonalization (Table 6).

## Discussion

Understanding the prevalence, predictors, and implications of burnout is vital for a profession that requires empathy and engagement, and especially so given its current state of crisis. Among academic family medicine faculty, the prevalence

**Table 2. Prevalence of emotional exhaustion, depersonalization and personal accomplishment scores among family medicine faculty (n = 623).**

| Likelihood of burnout | Emotional Exhaustion N (%) | Depersonalization N (%) | Personal Accomplishment* N (%) |
|---|---|---|---|
| Low Score | 267 (42.9) | 427 (68.5) | 619 (99.4) |
| Moderate Score | 188 (30.2) | 139 (22.3) | 3 (0.5) |
| *High Score* | *168 (27.0)* | *57 (9.2)* | *1 (0.2)* |

*Personal accomplishment is scored in the opposite direction such that the low score range denotes a high level of personal accomplishment.

**Table 3. Potential predictors of emotional exhaustion among family medicine faculty (n = 623).**

| Potential Predictor Variables | Emotional Exhaustion (EE) | | P-value (χ2 test) | Odds Ratio | 95% CI |
|---|---|---|---|---|---|
| | High EE (n = 168) | Low/Moderate EE (455) | | | |
| **Faculty Ratings of Local Department** | | | | | |
| Rating of overall *support* for teaching, research, leadership, mentorship, and career (n = 623): | | | | | |
| Good/fair/poor | 71 (36.0) | 126 (64.0) | <0.001 | 1.91 | (1.32, 2.76) |
| Very good/excellent | 97 (22.8) | 329 (77.2) | | "---" | – |
| Rating of overall *recognition* of teaching, research, leadership, mentorship and career support (n = 623): | | | | | |
| Good/fair/poor | 95 (34.2) | 183 (65.8) | <0.001 | 1.93 | (1.35, 2.77) |
| Very good/excellent | 73 (21.2) | 272 (78.8) | | "---" | – |
| Rating of communication (n = 623): | | | | | |
| Good/fair/poor | 88 (35.3) | 161 (64.7) | <0.01 | 2.01 | (1.40, 2.88) |
| Very good/excellent | 80 (21.4) | 294 (78.6) | | "---" | – |
| Rating of leadership (n = 623): | | | | | |
| Good/fair/poor | 53 (32.9) | 108 (67.1) | 0.048 | 1.48 | (1.00, 2.19) |
| Very good/excellent | 115 (24.9) | 347 (75.1) | | "---" | – |
| Rating of mission, vision and values (n = 623): | | | | | |
| Good/fair/poor | 95 (35.1) | 176 (64.9) | <0.001 | 2.06 | (1.44, 2.95) |
| Very good/excellent | 73 (20.7) | 279 (79.3) | | "---" | – |
| Rating of workload and practice (n = 623): | | | | | |
| Good/fair/poor | 81 (39.1) | 126 (60.9) | <0.001 | 2.43 | (1.69, 3.50) |
| Very good/excellent | 87 (20.9) | 329 (79.1) | | "---" | – |
| Rating of teamwork (n = 623): | | | | | |
| Good/fair/poor | 84 (39.1) | 131 (60.9) | <0.001 | 2.47 | (1.72, 3.56) |
| Very good/excellent | 84 (20.6) | 324 (79.4) | | "---" | – |
| Rating of physician involvement in programs and planning (n = 623): | | | | | |
| Good/fair/poor | 98 (34.6) | 185 (65.4) | <0.001 | 2.04 | (1.43, 2.93) |
| Very good/excellent | 70 (20.6) | 270 (79.4) | | "---" | – |
| Rating of resource distribution for clinical work, teaching and research (n = 623): | | | | | |
| Good/fair/poor | 104 (32.5) | 216 (67.5) | 0.001 | 1.80 | (1.25, 2.58) |
| Very good/excellent | 64 (21.1) | 239 (78.9) | | "---" | – |
| Rating of remuneration (n = 623): | | | | | |
| Good/fair/poor | 108 (37.9) | 211 (66.1) | <0.001 | 2.08 | (1.44, 3.00) |
| Very good/excellent | 60 (19.7) | 244 (80.3) | | "---" | – |
| Rating of respect (n = 623): | | | | | |
| Good/fair/poor | 81 (37.9) | 133 (62.1) | <0.001 | 2.25 | (1.57, 3.24) |
| Very good/excellent | 87 (21.3) | 322 (78.7) | | "---" | – |
| **Faculty Ratings of Main Practice Setting** | | | | | |
| Rating of main practice setting with regards to infrastructure support (n = 623): | | | | | |
| Good/fair/poor | 39 (35.5) | 71 (64.5) | 0.027 | 1.64 | (1.06, 2.54) |
| Very good/excellent | 129 (25.1) | 384 (74.9) | | "---" | – |

*(Continued)*

**Table 3.** (Continued)

| Potential Predictor Variables | Emotional Exhaustion (EE) | | P-value (χ2 test) | Odds Ratio | 95% CI |
|---|---|---|---|---|---|
| | High EE (n = 168) | Low/Moderate EE (455) | | | |
| **Leadership and Mentorship Experiences** | | | | | |
| Rating of importance of barriers in taking on a leadership role (n = 623): | | | | | |
| Somewhat/very important | 93 (30.9) | 208 (69.1) | 0.033 | 1.47 | (1.03, 2.10) |
| Not at all/not very/neutral | 75 (23.3) | 247 (76.7) | | "---" | – |
| Rating of the overall quality of mentoring received (n = 623): | | | | | |
| Good/fair/poor | 95 (35.2) | 175 (64.8) | <0.001 | 2.08 | (1.45, 2.98) |
| Very good/excellent | 73 (20.7) | 280 (79.3) | | "---" | – |
| **Job Satisfaction** | | | | | |
| Rating of overall job satisfaction (n = 623): | | | | | |
| Very dissatisfied to satisfied | 146 (47.7) | 160 (52.3) | <0.001 | 12.24 | (7.51, 19.93) |
| Very satisfied | 22 (6.9) | 295 (93.1) | | "---" | – |
| Rating of the quality of local department as a place to work (n = 623): | | | | | |
| Good/fair/poor | 94 (42.2) | 129 (57.8) | <0.001 | 3.21 | (2.23, 4.63) |
| Very good/excellent | 74 (18.5) | 326 (81.5) | | "---" | – |
| Rating of the likelihood to recommend local department as a place to work (n = 623): | | | | | |
| Other response | 108 (39.6) | 165 (60.4) | <0.001 | 3.16 | (2.19, 4.58) |
| Very likely | 60 (17.1) | 290 (82.9) | | "---" | – |
| Rating of the likelihood to leave local department in the next 5 years (n = 623): | | | | | |
| Somewhat/very likely | 57 (44.5) | 71 (55.5) | <0.001 | 2.78 | (1.85, 4.17) |
| Other response | 111 (22.4) | 384 (77.6) | | "---" | – |
| **Health Status Variables** | | | | | |
| Self rated health status (n = 623): | | | | | |
| Poor/fair/good | 43 (47.8) | 47 (52.2) | <0.001 | 2.98 | (1.88, 4.72) |
| Very good/excellent | 125 (23.5) | 407 (76.5) | | "---" | – |
| Number of days *physical health* was not good in the last month (n = 622): | | | | | |
| 1–30 days | 88 (33.3) | 176 (66.7) | 0.002 | 1.74 | (1.22, 2.48) |
| 0 days | 80 (22.3) | 278 (77.7) | | "---" | – |
| Number of days *mental health* was not good in the last month (n = 622): | | | | | |
| 1–30 days | 134 (38.3) | 216 (61.7) | <0.001 | 4.34 | (2.86, 6.60) |
| 0 days | 34 (12.5) | 238 (87.5) | | "---" | – |
| Number of days poor physical or mental health prevented doing usual activities (n = 622): | | | | | |
| 1–30 days | 70 (40.2) | 104 (59.8) | <0.001 | 2.40 | (1.65, 3.50) |
| 0 days | 98 (21.9) | 350 (35.4) | | "---" | – |
| Self rated stress at *work* in the past year (n = 622): | | | | | |
| Extremely/quite stressful | 84 (64.6) | 46 (35.4) | <0.001 | 8.85 | (5.78, 13.70) |
| Other | 84 (17.1) | 408 (82.9) | | "---" | – |

*(Continued)*

**Table 3.** (Continued)

| Potential Predictor Variables | Emotional Exhaustion (EE) | | P-value (χ2 test) | Odds Ratio | 95% CI |
|---|---|---|---|---|---|
| | High EE (n = 168) | Low/Moderate EE (455) | | | |
| Self rated stress *in life* in the past year (n = 622): | | | | | |
| Extremely/quite stressful | 74 (66.1) | 38 (33.9) | <0.001 | 8.62 | (5.49, 13.51) |
| Other | 94 (18.4) | 416 (81.6) | | "---" | – |
| **Demographic and Practice Variables** | | | | | |
| Rating of the stress related to on-call responsibilities: (n = 539): | | | | | |
| Extremely/very stressful | 36 (43.9) | 46 (56.4) | <0.001 | 2.25 | (1.39, 3.65) |
| Other | 118 (25.8) | 339 (74.2) | | "---" | – |
| Number of hours worked per week, excluding on-call (n = 622): | | | | | |
| 50 or more hours | 52 (38.8) | 82 (61.2) | <0.001 | 2.03 | (1.36, 3.05) |
| Less than 50 hours | 116 (23.8) | 372 (76.2) | | "---" | – |
| Faculty member marital status (n = 616): | | | | | |
| Other | 29 (36.3) | 51 (63.7) | 0.049 | 1.64 | (1.00, 2.69) |
| Married/living with partner | 138 (25.7) | 398 (74.3) | | "---" | – |

of high emotional exhaustion was 27% and high depersonalization was 9%, even with almost universal high personal accomplishment (99%).

A metanalysis demonstrated immense variability in the prevalence of burnout (0%−80%), as well as variability for each of the MBI subscales (emotional exhaustion: 0–86.2%; depersonalization: 0–89.9%; and personal accomplishment: 0–87.1%) partly due to the inconsistent and unclear use of the term burnout [10,27]. This complicates interpretation and comparisons across studies [28–30].

In our study, the prevalence estimates of emotional exhaustion and depersonalization fall in the lower range of published studies, raising the possibility that these 2011, pre-pandemic, levels are indicative of "unavoidable" burnout inherent in physician work [11]. The explosion in the burnout literature and the current health workforce crisis suggest that burnout prevalence is increasing over time, and that current levels of burnout pose a greater challenge to physicians. Even when adjusted for hours worked, studies illustrate increasing levels of physician burnout, and higher levels of burnout among physicians compared to the general population [18,31]. Despite the wide variability in prevalence studies, all report burnout as a persistent issue impacting the physician workforce and potentially patient outcomes. Emphasizing the importance of a nuanced understanding of avoidable and unavoidable burnout predictors [11], a framework of avoidable and unavoidable burnout among family physicians further underpins the necessity for a detailed understanding of modifiable and unmodifiable predictors.

## Predictors of emotional exhaustion

The four independent predictors of emotional exhaustion – lower ratings of job satisfaction, poorer ratings of workplace quality, working ≥50 hours/week, and poorer ratings of health status – are supported in the literature and could be used to inform system-level changes, program development, and workplace policies that mitigate avoidable burnout [16,17,19,32].

We found that low job satisfaction is a strong predictor for both emotional exhaustion and depersonalization. Job satisfaction is a multidimensional construct that includes both unmodifiable factors (born in Canada) and modifiable factors [23]. Numerous other studies have identified time pressures, chaos, lack of work control, poor career fit, and loss of

**Table 4. Potential predictors of depersonalization among family medicine faculty (n = 623).**

| Potential Predictor Variables | Depersonalization | | P-value (χ2 test) | Odds Ratio | 95% CI |
|---|---|---|---|---|---|
| | High (n = 57) | Low/Moder-ate (n = 566) | | | |
| **Faculty Ratings of Local Department** | | | | | |
| Rating of overall *support* for teaching, research, leadership, mentorship, and career (n = 623): | | | | | |
| Good/fair/poor | 26 (13.2) | 171 (86.8) | 0.017 | 1.94 | (1.12, 3.36) |
| Very good/excellent | 31 (7.3) | 395 (92.7) | | "---" | – |
| Rating of communication (n = 623): | | | | | |
| Good/fair/poor | 31 (12.4) | 218 (87.6) | 0.020 | 1.90 | (1.10, 3.29) |
| Very good/excellent | 26 (7.0) | 348 (93.0) | | "---" | – |
| Rating of workload and practice (n = 623): | | | | | |
| Good/fair/poor | 27 (13.0) | 180 (87.0) | 0.017 | 1.93 | (1.11, 3.34) |
| Very good/excellent | 30 (7.2) | 386 (92.8) | | "---" | – |
| Rating of teamwork (n = 623): | | | | | |
| Good/fair/poor | 28 (13.0) | 187 (87.0) | 0.015 | 1.96 | (1.13, 3.39) |
| Very good/excellent | 29 (7.1) | 379 (92.9) | | "---" | – |
| Rating of physician involvement in programs and planning (n = 623): | | | | | |
| Good/fair/poor | 34 (12.0) | 249 (88.0) | 0.024 | 1.88 | (1.08, 3.28) |
| Very good/excellent | 23 (6.8) | 317 (93.2) | | "---" | – |
| Rating of resource distribution for clinical work, teaching and research (n = 623): | | | | | |
| Good/fair/poor | 38 (11.9) | 282 (88.1) | 0.015 | 2.01 | (1.13, 3.58) |
| Very good/excellent | 19 (6.3) | 284 (93.7) | | "---" | – |
| Rating of remuneration (n = 623): | | | | | |
| Good/fair/poor | 37 (11.6) | 282 (88.4) | 0.030 | 1.86 | (1.06, 3.29) |
| Very good/excellent | 20 (6.6) | 284 (93.4) | | "---" | – |
| Rating of respect (n = 623): | | | | | |
| Good/fair/poor | 27 (12.6) | 187 (87.4) | 0.030 | 1.82 | (1.05, 3.16) |
| Very good/excellent | 30 (7.3) | 379 (92.7) | | "---" | – |
| **Leadership and Mentorship Experiences** | | | | | |
| Rating of the overall quality of mentoring received (n = 623): | | | | | |
| Good/fair/poor | 33 (12.2) | 237 (87.8) | 0.020 | 1.91 | (1.10, 3.31) |
| Very good/excellent | 24 (6.8) | 329 (93.2) | | "---" | – |
| **Job Satisfaction** | | | | | |
| Rating of overall job satisfaction (n = 623): | | | | | |
| Very dissatisfied to satisfied | 47 (15.4) | 259 (84.6) | <0.001 | 5.59 | (2.76, 11.24) |
| Very satisfied | 10 (3.2) | 307 (96.8) | | "---" | – |
| Rating of the quality of local department as a place to work (n = 623): | | | | | |
| Good/fair/poor | 28 (12.6) | 195 (87.4) | 0.028 | 1.84 | (1.06, 3.18) |
| Very good/excellent | 29 (7.2) | 371 (92.8) | | "----" | – |
| Rating of the likelihood to recommend local department as a place to work (n = 623): | | | | | |
| Other response | 40 (14.7) | 233 (85.3) | <0.001 | 3.36 | (1.86, 6.08) |
| Very likely | 17 (4.9) | 333 (95.1) | | "----" | – |
| Rating of the likelihood to leave local department in the next 5 years (n = 623): | | | | | |
| Somewhat/very likely | 21 (16.4) | 107 (83.6) | 0.001 | 2.50 | (1.40, 4.46) |
| Other response | 36 (7.3) | 459 (92.7) | | "---" | – |

*(Continued)*

**Table 4.** (Continued)

| Potential Predictor Variables | Depersonalization | | P-value (χ2 test) | Odds Ratio | 95% CI |
|---|---|---|---|---|---|
| | High (n = 57) | Low/Moder-ate (n = 566) | | | |
| **Health Status Variables** | | | | | |
| Self rated health status (n = 623): | | | | | |
| Poor/fair/good | 18 (20.0) | 72 (80.0) | <0.001 | 3.16 | (1.72, 5.82) |
| Very good/excellent | 39 (7.3) | 493 (92.7) | | "---" | – |
| Self rated stress at *work* in the past year (n = 622): | | | | | |
| Extremely/quite stressful | 32 (24.6) | 98 (75.4) | <0.001 | 6.10 | (3.46, 10.75) |
| Other | 25 (5.1) | 467 (94.9) | | "---" | – |
| Self rated stress *in life* in the past year (n = 622): | | | | | |
| Extremely/quite stressful | 21 (18.3) | 94 (81.7) | <0.001 | 2.99 | (1.66, 5.41) |
| Other | 34 (6.9) | 456 (93.1) | | "---" | – |
| **Demographic and Practice Variables** | | | | | |
| Length of time licensed for independent practice (n = 605): | | | | | |
| 5 years or less | 21 (18.3) | 94 (81.7) | <0.001 | 3.00 | (1.67, 5.39) |
| 6 or more years | 34 (6.9) | 456 (93.1) | | "---" | – |
| Faculty member's age (n = 604): | | | | | |
| Less than 50 years of age | 43 (11.7) | 323 (88.3) | 0.009 | 2.30 | (1.21, 4.38) |
| 50 years of age or older | 13 (5.5) | 225 (94.5) | | "---" | – |

meaning in work due to high administrative burden as contributors to lower ratings of job satisfaction, burnout, and intent to leave practice [17]. Based on our identified predictors, efforts aimed at improving overall job satisfaction by leveraging teamwork and mentorship opportunities would help address burnout.

Poor rating of workplace quality was also identified as a predictor of emotional exhaustion. Workplace quality is a composite variable based on ratings of the following three items: being a comfortable place to practice, being free from operational and bureaucratic difficulties; and being a fun and positive place to work. Those who did not rate their workplace highly were more likely to be emotionally exhausted. Programs such as the American Medical Association's Steps Forward Program to create "Joy in Medicine" provide a framework to address workplace quality by highlighting three crucial steps: culture change, addressing clinical inefficiencies, and initiatives to enhance health provider resilience [33,34]. It is notable that we did not identify renumeration (either low or high) as a predictor of burnout in our multivariate analysis. Pay increases and financial incentives often appear to be the panacea for improving workplace quality. However, providing financial incentives alone without addressing workplace quality has been shown to contribute to depersonalized care and hamper practice [35]. Intrinsic factors that support well-being include autonomy with respect to time spent in patient care, competence to exercise clinical judgement, relatedness to patients and the organization, and noted appreciation for academic and administrative duties [36,37]. In challenging fiscal times, leaders and organizations can leverage an understanding of workplace quality as a predictor to actively mitigate burnout. The connection between quality of the workplace culture, values, leadership, and physician well-being is well documented in the literature [38,39].

Respondents who reported working ≥50 hours/week (excluding on-call) were more likely to have high emotional exhaustion than those working <50 hours/week. Long work hours, high workload, and overnight call have been associated with burnout (8). Beyond hours worked, physicians who spent at least 20% of their time on tasks they found meaningful were at lower risk of burnout [38,40]. Actively addressing modifiable predictors of emotional exhaustion including job satisfaction, workplace quality, hours spent at work, and meaningful work could mitigate avoidable burnout, and provide essential levers for leaders and institutions.

**Table 5. Logistic regression of the most important predictors of emotional exhaustion among family medicine faculty (n = 622).**

| Predictors of Emotional Exhaustion | Adjusted Odds Ratio | 95% Confidence Interval |
|---|---|---|
| Rating of overall job satisfaction: | | |
| Very dissatisfied to satisfied | 10.21 | (6.19, 16.83) |
| Very satisfied | "---" | – |
| Rating of the quality of local department as a place to work:[3] | | |
| Good/fair/poor | 2.14 | (1.41, 3.24) |
| Very good/excellent | "---" | – |
| Number of hours worked per week, excluding on-call: | | |
| 50 or more hours | 1.93 | (1.20, 3.10) |
| Less than 50 hours | "---" | – |
| Self-rated health status: | | |
| Poor/fair/good | 1.88 | (1.10, 3.19) |
| Very good/excellent | "---" | – |

*Final Logistic Regression Model Statistics:*

Rho-square = .26 (pseudo $R^2$, values between 0.2 and 0.4 suggest a very good model fit).

Cox & Snell R-square = .241; Nagelkerke R-square = .350 (i.e., between 24.1% and 35.0% of variance is explained by this model).

Hosmer and Lemeshow Goodness-of-Fit test = 0.295 (values greater than 0.25 indicate good fit).

78.0% correctly classified.

Our study also highlights poor health status as a predictor of both emotional exhaustion and depersonalization. Specific health diagnoses associated with burnout are not identified in most studies, however the literature does describe an overlap between depression, psychiatric illness, and burnout [19,29,41]. Stigma of disclosing illness, mental health conditions, and addictions continues to predominate in physician culture. Fear of loss of licensure prevents many physicians from seeking care for treatable health issues [11]. Institutions could examine strategies, policies, and practices that reduce the stigma associated with reporting illness and incorporate workplace modifications for those affected. Strategies that delicately balance privacy and a fulsome understanding of health status as a burnout predictor may potentially provide workforce-sustaining improvements.

## Predictors of depersonalization

The four independent predictors of depersonalization in our study were: lower ratings of job satisfaction (discussed previously), shorter duration (<5 years) in practice, lower ratings of health status (discussed previously), and poorer ratings of mentorship received. Early career family physicians are at risk for burnout because transition to independent practice is a time of additional stress [42]. A steep practice management learning curve, misalignment in career fit, adapting to new practice sites, and new family responsibilities are potential contributors. Given that academic departments are the context for training future family physicians, research with recent graduates is an important area for further study. Burnout among family physicians may dissuade trainees from entering the discipline or pursuing comprehensive family medicine after graduation [43].

Our findings suggest that improving the modifiable factors of job satisfaction, health status when possible, and mentorship received may help reduce depersonalization for all family physicians and perhaps more potently for new graduates. The impact of high-quality mentorship is documented in the literature [24,44]. The College of Family Physicians of Canada has examined needs of Early Career Family Physicians (ECFPs) and identified gaps around mentorship related

**Table 6. Logistic regression of the most important predictors of depersonalization among family medicine faculty (n = 605).**

| Predictors of Depersonalization | Adjusted Odds Ratio | 95% Confidence Interval |
|---|---|---|
| Rating of overall job satisfaction | | |
| Very dissatisfied to satisfied | 4.71 | (2.22, 9.99) |
| Very satisfied | "---" | – |
| Length of time licensed for independent practice: | | |
| 5 years or less | 3.91 | (2.03, 7.51) |
| 6 or more years | "---" | – |
| Self-rated health status: | | |
| Poor/fair/good | 2.98 | (1.53, 5.88) |
| Very good/excellent | "---" | – |
| Rating of the overall quality of mentoring received: | | |
| Good/fair/poor | 1.92 | (1.04, 3.56) |
| Very good/excellent | "---" | – |

*Final Logistic Regression Model Statistics:*

Rho-square = .15 (pseudo $R^2$, values between 0.2 and 0.4 suggest a very good model fit).

Cox & Snell R-square = .088; Nagelkerke R-square = .193 (i.e., between 8.8% and 19.3% of variance is explained by this model).

Hosmer and Lemeshow Goodness-of-Fit test = 0.715 (values greater than 0.25 indicate good fit).

91.1% correctly classified.

to practice management issues, lack of awareness among ECFPs on how to connect with a mentor, and issues with sustaining mentor capacity [45,46]. Both national and provincial family medicine regulatory bodies have launched mentorship programs to address the needs of ECFPs to support this stage in the healthcare workforce [47]. Reflection on these predictors could provide trainees, family physicians, leaders, and departments of family medicine opportunities to mitigate avoidable burnout and create optimum recovery initiatives to address unavoidable burnout.

## Strengths and limitations

The strengths of this research include the comprehensive questionnaire, the rigorous approach to survey design and implementation, the high response rate, and the sequential application of bivariate analysis followed by multivariable analysis. The limitation that it was conducted at a single academic department of family medicine may be diminished given the large number of participants in multiple diverse sites, suggesting that the findings may be generalizable to many family medicine settings. Another limitation relates to the age of these data, which reflect a snapshot in time and may not represent the current situation. However, these historical data provide important information for addressing burnout among Canadian family physicians, an issue that is important today. A final limitation is that cross-sectional studies, although informative about associations, generally cannot prove causation.

## Conclusion

This study identified that 27% of academic family physicians self-reported high levels of emotional exhaustion and 9% reported high levels of depersonalization, despite 99% reporting high levels of personal accomplishment. Identifying independent predictors of emotional exhaustion and depersonalization point to practice- and systems-level interventions to mitigate these avoidable components of burnout. These data from 2011 provide relevant comparators for assessing the impact of subsequent healthcare system changes, including the COVID-19 pandemic, information and digital technology,

and declining numbers of family physicians. Ongoing assessments of the prevalence of burnout and its correlates are warranted. Recent changes in medical practice including the rise of artificial intelligence, the evolution of electronic medical records, and changes in health teams and practice models support a longitudinal examination of family physician burnout and the impact of these emergent factors. This study provides an opportunity to look back to plan forward.

## Acknowledgments

The authors wish to thank Dr. Lynn Wilson, Professor and Past Chair of the Department of Family & Community Medicine for her support of the Departmental Academic Leadership Task Force and this research, as well as the Peer Support Writing Group at Women's College Hospital..

## Author contributions

**Conceptualization:** Viola Antao, Paul Krueger, Christopher Meaney, Jeffrey C. Kwong, David White.

**Data curation:** Paul Krueger, Christopher Meaney.

**Formal analysis:** Viola Antao, Paul Krueger, Christopher Meaney, Jeffrey C. Kwong, David White.

**Investigation:** Viola Antao, Paul Krueger, Christopher Meaney, Jeffrey C. Kwong, David White.

**Methodology:** Viola Antao, Paul Krueger, Christopher Meaney, Jeffrey C. Kwong, David White.

**Project administration:** Viola Antao, Paul Krueger, Christopher Meaney, Jeffrey C. Kwong, David White.

**Validation:** Paul Krueger, Christopher Meaney.

**Visualization:** Viola Antao, Paul Krueger, Christopher Meaney, Jeffrey C. Kwong, David White.

**Writing – original draft:** Viola Antao, Paul Krueger, Christopher Meaney, Jeffrey C. Kwong, David White.

**Writing – review & editing:** Viola Antao, Paul Krueger, Christopher Meaney, Jeffrey C. Kwong, David White.

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
