## [Decision Letter · Decision Letter 0]

20 Nov 2025

Dear Dr. Antao,

We look forward to receiving your revised manuscript.

Kind regards,

Gholamheidar Teimori-Boghsani

Academic Editor

PLOS ONE

Reviewers' comments:

Reviewer's Responses to Questions

**Comments to the Author**

1. Is the manuscript technically sound, and do the data support the conclusions?

Reviewer #1: Yes

Reviewer #2: Partly

2. Has the statistical analysis been performed appropriately and rigorously?

Reviewer #1: Yes

Reviewer #2: No

3. Have the authors made all data underlying the findings in their manuscript fully available?

Reviewer #1: No

Reviewer #2: Yes

4. Is the manuscript presented in an intelligible fashion and written in standard English?

Reviewer #1: Yes

Reviewer #2: Yes

Reviewer #1: The authors conducted a survey using a validated tool to measure factors that may predict burnout among family physicians in a region of Canada. They obtained responses from over 60% of the target group.

The language throughout the draft is excellent, and I find the abstract consistent with the content presented. The statistical analyses appear reasonable, although this is outside my area of expertise. The manuscript is well-structured, and the authors are appropriately cautious in their conclusions.

However, I suggest omitting the final sentence—“This study provides an opportunity to look back to plan forward”—as it reads more like a slogan than a scientific conclusion.

There are a few minor spelling errors and some illogical use of punctuation, which may have occurred during the conversion from Word to PDF.

Major Concerns:

Data Age: The dataset is 14 years old, raising questions about its relevance to current predictors of burnout among family physicians in Canada. While the authors briefly acknowledge this issue, I believe it deserves more thorough reflection. For example, they could discuss how working conditions have evolved over the past 15 years, including the impact of social media, technological advancements, the integration of AI, and the potential for AI-generated clinical documentation—all of which may influence burnout risk.

Tool Relevance: How applicable is the burnout assessment tool today? For instance, Sullivan et al. (2025) found that 78% of healthcare workers’ responses fell outside the three dimensions measured by the Maslach Burnout Inventory (MBI), suggesting that the tool may no longer fully capture the modern experience of burnout.

Reference: Sullivan et al. (2025). Healthcare worker burnout: Rethinking the Maslach Burnout Inventory. Psychology, Health & Medicine. https://doi.org/10.1080/13548506.2025.2487949

Minor Concerns:

a. In the introduction, the authors state that 6.5 million Canadians lack access to primary care, and “yet” 100 family residency positions remain unfilled. These two statements seem disconnected. In the context of the entire country, 100 unfilled positions is not a particularly high number.

b. Line 16: “Among family physicians, the prevalence ranged from 25–60%.” It would be helpful to include comparative prevalence rates from other medical specialties for context.

Reviewer #2: Hello dear authors.

MS Id: PONE-D-25-29490

Title: Predictors of Burnout Among Academic Family Medicine Faculty: Looking Back to Plan Forward

Type: Research Article

Here are my recommendations about the mentioned MS:

Title:

• Looks good.

Abstract:

• Write subsections of method in abstract in a structured manner.

• Do not use abbreviations or symbols in abstract that did not describe previously.

• Summarize results and conclusions.

Introduction:

• Using the word “Burnout” frequently make readers confuse you can use adjectives instead of repeating the word “Burnout”.

Methodology:

• Period, population, and sampling method have to be mentioned in the method section.

• Dedicate a subsection for describing study variables, and determining the sample size.

• The tool's reliability has to be existed.

• The pilot study and the reliability process for the tools have not been mentioned.

• How do you determine cut-off points? Please explain it.

• Normality distribution of the data and the software which used for analyzing the data have to be mentioned.

Results:

• Provide a table for presenting sociodemographic characteristics of the participants.

• Line 85-88 have to be moved to the method.

• Write Comments for all of the tables.

• Write p values in the table 5 and 6.

Discussion:

• Looks good.

Conclusion:

• Looks good.

References:

• Change the reference style to the Vancouver style.

Figures and tables:

• No figure exists.

.

Reviewer #1: **Yes:** Eivind AakhusEivind AakhusEivind AakhusEivind Aakhus

Reviewer #2: No

---

## [Author Response · Author response to Decision Letter 1]

27 Dec 2025

Thank you to the Editor, and reviewer Eivind Aakhus and Reviewer #2, the authors appreciate your in depth review and comments. Please see the detailed response letter uploaded.

Kindest Regards,

Viola Antao on behalf of the authors

---

## [Decision Letter · Decision Letter 1]

25 Feb 2026

Predictors of Burnout Among Academic Family Medicine Faculty:Looking Back to Plan Forward

PONE-D-25-29490R1

Dear Dr. Antao,

We’re pleased to inform you that your manuscript has been judged scientifically suitable for publication and will be formally accepted for publication once it meets all outstanding technical requirements.

Kind regards,

Gholamheidar Teimori-Boghsani

Academic Editor

PLOS One

Additional Editor Comments (optional):

Reviewers' comments:

Reviewer's Responses to Questions

**Comments to the Author**

Reviewer #1: All comments have been addressed

2. Is the manuscript technically sound, and do the data support the conclusions?

Reviewer #1: Yes

3. Has the statistical analysis been performed appropriately and rigorously?

Reviewer #1: Yes

4. Have the authors made all data underlying the findings in their manuscript fully available?

Reviewer #1: No

5. Is the manuscript presented in an intelligible fashion and written in standard English?

Reviewer #1: Yes

Reviewer #1: I think the authors have addressed my concerns regarding a potential for using an outdated tool to measure burn out. I also agree with the authors that p is not necessary, as long as CI is presented, but this should be discussed with the second reviewer. If I'm the only reviewer to assess this second version p calculations could better be omitted in the tables.

.

Reviewer #1: **Yes:** Eivind AakhusEivind AakhusEivind AakhusEivind Aakhus

---

## [Editor Report · Acceptance letter]

PONE-D-25-29490R1

PLOS One

Dear Dr. Antao,

I'm pleased to inform you that your manuscript has been deemed suitable for publication in PLOS One. Congratulations! Your manuscript is now being handed over to our production team.

Kind regards,

on behalf of

Dr. Gholamheidar Teimori-Boghsani

Academic Editor

PLOS One